# END-TO-END ADVERSARIAL TEXT-TO-SPEECH

**Jeff Donahue**, **Sander Dieleman**, **Mikołaj Bińkowski**, **Erich Elsen**, **Karen Simonyan**[*]
DeepMind
`{jeffdonahue,sedielem,binek,eriche,simonyan}@google.com`

## ABSTRACT

Modern text-to-speech synthesis pipelines typically involve multiple processing stages, each of which is designed or learnt independently from the rest. In this work, we take on the challenging task of learning to synthesise speech from normalised text or phonemes in an end-to-end manner, resulting in models which operate directly on character or phoneme input sequences and produce raw speech audio outputs. Our proposed generator is feed-forward and thus efficient for both training and inference, using a differentiable alignment scheme based on token length prediction. It learns to produce high fidelity audio through a combination of adversarial feedback and prediction losses constraining the generated audio to roughly match the ground truth in terms of its total duration and mel-spectrogram. To allow the model to capture temporal variation in the generated audio, we employ soft dynamic time warping in the spectrogram-based prediction loss. The resulting model achieves a mean opinion score exceeding 4 on a 5 point scale, which is comparable to the state-of-the-art models relying on multi-stage training and additional supervision.[1]

## 1 INTRODUCTION

A text-to-speech (TTS) system processes natural language text inputs to produce synthetic human-like speech outputs. Typical TTS pipelines consist of a number of stages trained or designed independently – e.g. text normalisation, aligned linguistic featurisation, mel-spectrogram synthesis, and raw audio waveform synthesis (Taylor, 2009). Although these pipelines have proven capable of realistic and high-fidelity speech synthesis and enjoy wide real-world use today, these modular approaches come with a number of drawbacks. They often require supervision at each stage, in some cases necessitating expensive "ground truth" annotations to guide the outputs of each stage, and sequential training of the stages. Further, they are unable to reap the full potential rewards of data-driven "end-to-end" learning widely observed in a number of prediction and synthesis task domains across machine learning.

In this work, we aim to simplify the TTS pipeline and take on the challenging task of synthesising speech from text or phonemes in an end-to-end manner. We propose *EATS* – End-to-end Adversarial Text-to-Speech – generative models for TTS trained adversarially (Goodfellow et al., 2014) that operate on either pure text or raw (temporally unaligned) phoneme input sequences, and produce raw speech waveforms as output. These models eliminate the typical intermediate bottlenecks present in most state-of-the-art TTS engines by maintaining learnt intermediate feature representations throughout the network.

Our speech synthesis models are composed of two high-level submodules, detailed in Section 2. An *aligner* processes the raw input sequence and produces relatively low-frequency (200 Hz) aligned features in its own learnt, abstract feature space. The features output by the aligner may be thought of as taking the place of the earlier stages of typical TTS pipelines – e.g., temporally aligned mel-spectrograms or linguistic features. These features are then input to the *decoder* which upsamples the features from the aligner by 1D convolutions to produce 24 kHz audio waveforms.

By carefully designing the aligner and guiding training by a combination of adversarial feedback and domain-specific loss functions, we demonstrate that a TTS system can be learnt nearly end-to-end,

---

[*]Equal contribution. First author determined by coin toss.
[1] Listen to our model reading this abstract at: `https://deepmind.com/research/publications/End-to-End-Adversarial-Text-to-Speech`

resulting in high-fidelity natural-sounding speech approaching the state-of-the-art TTS systems. Our main contributions include:

- A fully differentiable and efficient feed-forward aligner architecture that predicts the duration of each input token and produces an audio-aligned representation.

- The use of flexible dynamic time warping-based prediction losses to enforce alignment with input conditioning while allowing the model to capture the variability of timing in human speech.

- An overall system achieving a mean opinion score of $4.083$, approaching the state of the art from models trained using richer supervisory signals.

## 2 METHOD

Our goal is to learn a neural network (the generator) which maps an input sequence of characters or phonemes to raw audio at 24 kHz. Beyond the vastly different lengths of the input and output signals, this task is also challenging because the input and output are not aligned, i.e. it is not known beforehand which output tokens each input token will correspond to. To address these challenges, we divide the generator into two blocks: (i) the aligner, which maps the unaligned input sequence to a representation which is aligned with the output, but has a lower sample rate of 200 Hz; and (ii) the decoder, which upsamples the aligner's output to the full audio frequency. The entire generator architecture is differentiable, and is trained end to end. Importantly, it is also a feed-forward convolutional network, which makes it well-suited for applications where fast batched inference is important: our EATS implementation generates speech at a speed of $200\times$ realtime on a single NVIDIA V100 GPU (see Appendix A and Table 3 for details). It is illustrated in Figure 1.

The generator is inspired by GAN-TTS (Bińkowski et al., 2020), a text-to-speech generative adversarial network operating on aligned linguistic features. We employ the GAN-TTS generator as the decoder in our model, but instead of upsampling pre-computed linguistic features, its input comes from the aligner block. We make it speaker-conditional by feeding in a speaker embedding $\mathbf{s}$ alongside the latent vector $\mathbf{z}$, to enable training on a larger dataset with recordings from multiple speakers. We also adopt the multiple random window discriminators (RWDs) from GAN-TTS, which have been proven effective for adversarial raw waveform modelling, and we preprocess real audio input by applying a simple $\mu$-law transform. Hence, the generator is trained to produce audio in the $\mu$-law domain and we apply the inverse transformation to its outputs when sampling.

The loss function we use to train the generator is as follows:

$$\mathcal{L}_G = \mathcal{L}_{G,\text{adv}} + \lambda_{\text{pred}} \cdot \mathcal{L}''_{\text{pred}} + \lambda_{\text{length}} \cdot \mathcal{L}_{\text{length}}, \tag{1}$$

where $\mathcal{L}_{G,\text{adv}}$ is the adversarial loss, linear in the discriminators' outputs, paired with the hinge loss (Lim & Ye, 2017; Tran et al., 2017) used as the discriminators' objective, as used in GAN-TTS (Bińkowski et al., 2020). The use of an adversarial (Goodfellow et al., 2014) loss is an advantage of our approach, as this setup allows for efficient feed-forward training and inference, and such losses tend to be mode-seeking in practice, a useful behaviour in a strongly conditioned setting where realism is an important design goal, as in the case of text-to-speech. In the remainder of this section, we describe the aligner network and the auxiliary prediction ($\mathcal{L}''_{\text{pred}}$) and length ($\mathcal{L}_{\text{length}}$) losses in detail, and recap the components which were adopted from GAN-TTS.

### 2.1 ALIGNER

Given a token sequence $\mathbf{x} = (x_1, \ldots, x_N)$ of length $N$, we first compute token representations $\mathbf{h} = f(\mathbf{x}, \mathbf{z}, \mathbf{s})$, where $f$ is a stack of dilated convolutions (van den Oord et al., 2016) interspersed with batch normalisation (Ioffe & Szegedy, 2015) and ReLU activations. The latents $\mathbf{z}$ and speaker embedding $\mathbf{s}$ modulate the scale and shift parameters of the batch normalisation layers (Dumoulin et al., 2017; De Vries et al., 2017). We then predict the length for each input token individually: $l_n = g(h_n, \mathbf{z}, \mathbf{s})$, where $g$ is an MLP. We use a ReLU nonlinearity at the output to ensure that the predicted lengths are non-negative. We can then find the predicted token end positions as a cumulative sum of the token lengths: $e_n = \sum_{m=1}^{n} l_m$, and the token centre positions as $c_n = e_n - \frac{1}{2}l_n$. Based on these predicted positions, we can interpolate the token representations into an audio-aligned representation at 200 Hz, $\mathbf{a} = (a_1, \ldots, a_S)$, where $S = \lceil e_N \rceil$ is the total number of output time steps. To compute $a_t$, we obtain interpolation weights for the token representations $h_n$ using a softmax over

the squared distance between $t$ and $c_n$, scaled by a temperature parameter $\sigma^2$, which we set to 10.0 (i.e. a Gaussian kernel):

$$w_t^n = \frac{\exp\left(-\sigma^{-2}(t - c_n)^2\right)}{\sum_{m=1}^{N} \exp\left(-\sigma^{-2}(t - c_m)^2\right)}. \tag{2}$$

Using these weights, we can then compute $a_t = \sum_{n=1}^{N} w_t^n h_n$, which amounts to non-uniform interpolation. By predicting token lengths and obtaining positions using cumulative summation, instead of predicting positions directly, we implicitly enforce monotonicity of the alignment. Note that tokens which have a non-monotonic effect on prosody, such as punctuation, can still affect the entire utterance thanks to the stack of dilated convolutions $f$, whose receptive field is large enough to allow for propagation of information across the entire token sequence. The convolutions also ensure generalisation across different sequence lengths. Appendix B includes pseudocode for the aligner.

## 2.2 WINDOWED GENERATOR TRAINING

Training examples vary widely in length, from about 1 to 20 seconds. We cannot pad all sequences to a maximal length during training, as this would be wasteful and prohibitively expensive: 20 seconds of audio at 24 kHz correspond to 480,000 timesteps, which results in high memory requirements. Instead, we randomly extract a 2 second window from each example, which we will refer to as a *training window*, by uniformly sampling a random offset $\eta$. The aligner produces a 200 Hz audio-aligned representation for this window, which is then fed to the decoder (see Figure 1). Note that we only need to compute $a_t$ for time steps $t$ that fall within the sampled window, but we do have to compute the predicted token lengths $l_n$ for the entire input sequence. During evaluation, we simply produce the audio-aligned representation for the full utterance and run the decoder on it, which is possible because it is fully convolutional.

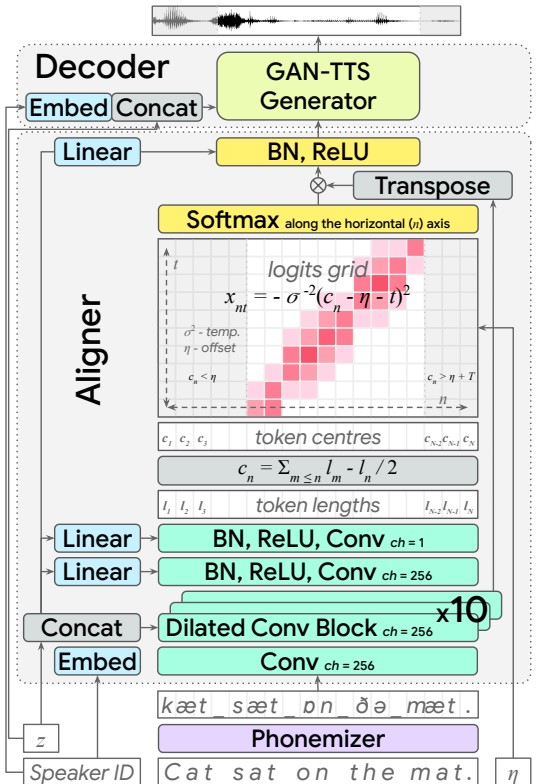

Figure 1: A diagram of the generator, including the monotonic interpolation-based aligner. $z$ and $ch$ denote the latent Gaussian vector and the number of output channels, respectively. During training, audio windows have a fixed length of 2 seconds and are generated from the conditioning text using random offsets $\eta$ and predicted phoneme lengths; the shaded areas in the logits grid and waveform are not synthesised. For inference (sampling), we set $\eta = 0$. In the No Phonemes ablation, the *phonemizer* is skipped and the character sequence is fed directly into the aligner.

## 2.3 ADVERSARIAL DISCRIMINATORS

**Random window discriminators.** We use an ensemble of random window discriminators (RWDs) adopted from GAN-TTS. Each RWD operates on audio fragments of different lengths, randomly sampled from the training window. We use five RWDs with window sizes 240, 480, 960, 1920 and 3600. This enables each RWD to operate at a different resolution. Note that 3600 samples at 24 kHz corresponds to 150 ms of audio, so all RWDs operate on short timescales. All RWDs in our model are *unconditional* with respect to text: they cannot access the text sequence or the aligner output. (GAN-TTS uses 10 RWDs, including 5 conditioned on linguistic features which we omit.) They are, however, conditioned on the speaker, via projection embedding (Miyato & Koyama, 2018).

**Spectrogram discriminator.** We use an additional discriminator which operates on the full training window in the spectrogram domain. We extract log-scaled mel-spectrograms from the audio signals and use the BigGAN-deep architecture (Brock et al., 2018), essentially treating the spectrograms as

images. The spectrogram discriminator also uses speaker identity through projection embedding. Details on the spectrogram discriminator architecture are included in Appendix C.

## 2.4 SPECTROGRAM PREDICTION LOSS

In preliminary experiments, we discovered that adversarial feedback is insufficient to learn alignment. At the start of training, the aligner does not produce an accurate alignment, so the information in the input tokens is incorrectly temporally distributed. This encourages the decoder to ignore the aligner output. The unconditional discriminators provide no useful learning signal to correct this. If we want to use conditional discriminators instead, we face a different problem: we do not have aligned ground truth. Conditional discriminators also need an aligner module, which cannot function correctly at the start of training, effectively turning them into unconditional discriminators. Although it should be possible in theory to train the discriminators' aligner modules adversarially, we find that this does not work in practice, and training gets stuck.

Instead, we propose to guide learning by using an explicit prediction loss in the spectrogram domain: we minimise the $L_1$ loss between the log-scaled mel-spectrograms of the generator output, and the corresponding ground truth training window. This helps training to take off, and renders conditional discriminators unnecessary, simplifying the model. Let $S_{\text{gen}}$ be the spectrogram of the generated audio, $S_{\text{gt}}$ the spectrogram of the corresponding ground truth, and $S[t, f]$ the log-scaled magnitude at time step $t$ and mel-frequency bin $f$. Then the prediction loss is:

$$\mathcal{L}_{\text{pred}} = \frac{1}{F} \sum_{t=1}^{T} \sum_{f=1}^{F} |S_{\text{gen}}[t, f] - S_{\text{gt}}[t, f]|. \tag{3}$$

$T$ and $F$ are the total number of time steps and mel-frequency bins respectively. Computing the prediction loss in the spectrogram domain, rather than the time domain, has the advantage of increased invariance to phase differences between the generated and ground truth signals, which are not perceptually salient. Seeing as the spectrogram extraction operation has several hyperparameters and its implementation is not standardised, we provide the code we used for this in Appendix D. We applied a small amount of jitter (by up to $\pm 60$ samples at 24 kHz) to the ground truth waveform before computing $S_{\text{gt}}$, which helped to reduce artifacts in the generated audio.

The inability to learn alignment from adversarial feedback alone is worth expanding on: likelihood-based autoregressive models have no issues learning alignment, because they are able to benefit from *teacher forcing* (Williams & Zipser, 1989) during training: the model is trained to perform next step prediction on each sequence step, given the preceding ground truth, and it is expected to infer alignment only one step at a time. This is not compatible with feed-forward adversarial models however, so the prediction loss is necessary to bootstrap alignment learning for our model.

Note that although we make use of mel-spectrograms for training in $\mathcal{L}_{\text{pred}}$ (and to compute the inputs for the spectrogram discriminator, Section 2.3), the generator itself does *not* produce spectrograms as part of the generation process. Rather, its outputs are raw waveforms, and we convert these waveforms to spectrograms only for training (backpropagating gradients through the waveform to mel-spectrogram conversion operation).

## 2.5 DYNAMIC TIME WARPING

The spectrogram prediction loss incorrectly assumes that token lengths are deterministic. We can relax the requirement that the generated and ground truth spectrograms are exactly aligned, by incorporating *dynamic time warping* (DTW) (Sakoe, 1971; Sakoe & Chiba, 1978). We calculate the prediction loss by iteratively finding a minimal-cost alignment path $p$ between the generated and target spectrograms, $S_{\text{gen}}$ and $S_{\text{gt}}$. We start at the first time step in both spectrograms: $p_{\text{gen},1} = p_{\text{gt},1} = 1$. At each iteration $k$, we take one of three possible actions:

1. go to the next time step in both $S_{\text{gen}}, S_{\text{gt}}$: $p_{\text{gen},k+1} = p_{\text{gen},k} + 1$, $p_{\text{gt},k+1} = p_{\text{gt},k} + 1$;
2. go to the next time step in $S_{\text{gt}}$ only: $p_{\text{gen},k+1} = p_{\text{gen},k}$, $p_{\text{gt},k+1} = p_{\text{gt},k} + 1$;
3. go to the next time step in $S_{\text{gen}}$ only: $p_{\text{gen},k+1} = p_{\text{gen},k} + 1$, $p_{\text{gt},k+1} = p_{\text{gt},k}$.

The resulting path is $p = \langle (p_{\text{gen},1}, p_{\text{gt},1}), \dots, (p_{\text{gen},K_p}, p_{\text{gt},K_p}) \rangle$, where $K_p$ is the length. Each action is assigned a cost based on the $L_1$ distance between $S_{\text{gen}}[p_{\text{gen},k}]$ and $S_{\text{gt}}[p_{\text{gt},k}]$, and a warp penalty $w$ which is incurred if we choose not to advance both spectrograms in lockstep (i.e. we are

*warping* the spectrogram by taking action 2 or 3; we use $w = 1.0$). The warp penalty thus encourages alignment paths that do not deviate too far from the identity alignment. Let $\delta_k$ be an indicator which is 1 for iterations where warping occurs, and 0 otherwise. Then the total path cost $c_p$ is:

$$c_p = \sum_{k=1}^{K_p} \left( w \cdot \delta_k + \frac{1}{F} \sum_{f=1}^{F} |S_{\text{gen}}[p_{\text{gen},k}, f] - S_{\text{gt}}[p_{\text{gt},k}, f]| \right). \qquad (4)$$

$K_p$ depends on the degree of warping ($T \leq K_p \leq 2T - 1$). The DTW prediction loss is then:

$$\mathcal{L}'_{\text{pred}} = \min_{p \in \mathcal{P}} c_p, \qquad (5)$$

where $\mathcal{P}$ is the set of all valid paths. $p \in \mathcal{P}$ only when $p_{\text{gen},1} = p_{\text{gt},1} = 1$ and $p_{\text{gen},K_p} = p_{\text{gt},K_p} = T$, i.e. the first and last timesteps of the spectrograms are aligned. To find the minimum, we use dynamic programming. Figure 2 shows a diagram of an optimal alignment path between two sequences.

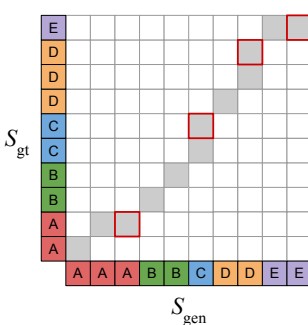

DTW is differentiable, but the minimum across all paths makes optimisation difficult, because the gradient is propagated only through the minimal path. We use a soft version of DTW instead (Cuturi & Blondel, 2017), which replaces the minimum with the soft minimum:

$$\mathcal{L}''_{\text{pred}} = -\tau \cdot \log \sum_{p \in \mathcal{P}} \exp\left(-\frac{c_p}{\tau}\right), \qquad (6)$$

where $\tau = 0.01$ is a temperature parameter and the loss scale factor $\lambda_{\text{pred}} = 1.0$. Note that the minimum operation is recovered by letting $\tau \to 0$. The resulting loss is a weighted aggregated cost across all paths, enabling gradient propagation through all feasible paths. This creates a trade-off: a higher $\tau$ makes optimisation easier, but the resulting loss less accurately reflects the minimal path cost. Pseudocode for the soft DTW procedure is provided in Appendix E.

Figure 2: Dynamic time warping between two sequences finds a minimal-cost alignment path. Positions where warping occurs are marked with a border.

By relaxing alignment in the prediction loss, the generator can produce waveforms that are not exactly aligned, without being heavily penalised for it. This creates a synergy with the adversarial loss: instead of working against each other because of the rigidity of the prediction loss, the losses now cooperate to reward realistic audio generation with stochastic alignment. Note that the prediction loss is computed on a training window, and not on full length utterances, so we still assume that the start and end points of the windows are exactly aligned. While this might be incorrect, it does not seem to be much of a problem in practice.

## 2.6 ALIGNER LENGTH LOSS

To ensure that the model produces realistic token length predictions, we add a loss which encourages the predicted utterance length to be close to the ground truth length. This length is found by summing all token length predictions. Let $L$ be the the number of time steps in the training utterance at 200 Hz, $l_n$ the predicted length of the $n$th token, and $N$ the number of tokens, then the length loss is:

$$\mathcal{L}_{\text{length}} = \frac{1}{2} \left( L - \sum_{n=1}^{N} l_n \right)^2. \qquad (7)$$

We use a scale factor $\lambda_{\text{length}} = 0.1$. Note that we cannot match the predicted lengths $l_n$ to the ground truth lengths individually, because the latter are not available.

## 2.7 TEXT PRE-PROCESSING

Although our model works well with character input, we find that sample quality improves significantly using phoneme input instead. This is not too surprising, given the heterogeneous way in which spellings map to phonemes, particularly in the English language. Many character sequences also have special pronunciations, such as numbers, dates, units of measurement and website domains, and a very large training dataset would be required for the model to learn to pronounce these correctly. Text normalisation (Zhang et al., 2019) can be applied beforehand to spell out these sequences as they are typically pronounced (e.g., *1976* could become *nineteen seventy six*), potentially followed by conversion to phonemes. We use an open source tool, *phonemizer* (Bernard, 2020), which performs partial normalisation and phonemisation (see Appendix F). Finally, whether we train on text or phoneme input sequences, we pre- and post-pad the sequence with a special silence token (for training and inference), to allow the aligner to account for silence at the beginning and end of each utterance.

## 3 RELATED WORK

Speech generation saw significant quality improvements once treating it as a generative modelling problem became the norm (Zen et al., 2009; van den Oord et al., 2016). Likelihood-based approaches dominate, but generative adversarial networks (GANs) (Goodfellow et al., 2014) have been making significant inroads recently. A common thread through most of the literature is a separation of the speech generation process into multiple stages: coarse-grained temporally aligned intermediate representations, such as mel-spectrograms, are used to divide the task into more manageable sub-problems. Many works focus exclusively on either spectrogram generation or vocoding (generating a waveform from a spectrogram). Our work is different in this respect, and we will point out which stages of the generation process are addressed by each model. In Appendix J, Table 6 we compare these methods in terms of the inputs and outputs to each stage of their pipelines.

Initially, most likelihood-based models for TTS were autoregressive (van den Oord et al., 2016; Mehri et al., 2017; Arik et al., 2017), which means that there is a sequential dependency between subsequent time steps of the produced output signal. That makes these models impractical for real-time use, although this can be addressed with careful engineering (Kalchbrenner et al., 2018; Valin & Skoglund, 2019). More recently, flow-based models (Papamakarios et al., 2019) have been explored as a feed-forward alternative that enables fast inference (without sequential dependencies). These can either be trained directly using maximum likelihood (Prenger et al., 2019; Kim et al., 2019; Ping et al., 2019b), or through distillation from an autoregressive model (van den Oord et al., 2018; Ping et al., 2019a). All of these models produce waveforms conditioned on an intermediate representation: either spectrograms or "linguistic features", which contain temporally-aligned high-level information about the speech signal. Spectrogram-conditioned waveform models are often referred to as *vocoders*.

A growing body of work has applied GAN (Goodfellow et al., 2014) variants to speech synthesis (Donahue et al., 2019). An important advantage of adversarial losses for TTS is a focus on realism over diversity; the latter is less important in this setting. This enables a more efficient use of capacity compared to models trained with maximum likelihood. MelGAN (Kumar et al., 2019) and Parallel WaveGAN (Yamamoto et al., 2020) are adversarial vocoders, producing raw waveforms from mel-spectrograms. Neekhara et al. (2019) predict magnitude spectrograms from mel-spectrograms. Most directly related to our work is GAN-TTS (Bińkowski et al., 2020), which produces waveforms conditioned on aligned linguistic features, and we build upon that work.

Another important line of work covers spectrogram generation from text. Such models rely on a vocoder to convert the spectrograms into waveforms (for which one of the previously mentioned models could be used, or a traditional spectrogram inversion technique (Griffin & Lim, 1984)). Tacotron 1 & 2 (Wang et al., 2017; Shen et al., 2018), Deep Voice 2 & 3 (Gibiansky et al., 2017; Ping et al., 2018), TransformerTTS (Li et al., 2019), Flowtron (Valle et al., 2020), and VoiceLoop (Taigman et al., 2017) are autoregressive models that generate spectrograms or vocoder features frame by frame. Guo et al. (2019) suggest using an adversarial loss to reduce exposure bias (Bengio et al., 2015; Ranzato et al., 2016) in such models. MelNet (Vasquez & Lewis, 2019) is autoregressive over both time and frequency. ParaNet (Peng et al., 2019) and FastSpeech (Ren et al., 2019) are non-autoregressive, but they require distillation (Hinton et al., 2015) from an autoregressive model. Recent flow-based approaches Flow-TTS (Miao et al., 2020) and Glow-TTS (Kim et al., 2020) are feed-forward without requiring distillation. Most spectrogram generation models require training of a custom vocoder model on generated spectrograms, because their predictions are imperfect and the vocoder needs to be able to compensate for this[2]. Note that some of these works also propose new vocoder architectures in tandem with spectrogram generation models.

Unlike all of the aforementioned methods, as highlighted in Appendix J, Table 6, our model is a single feed-forward neural network, trained end-to-end in a single stage, which produces waveforms given character or phoneme sequences, and learns to align without additional supervision from auxiliary sources (e.g. temporally aligned linguistic features from an external model) or teacher forcing. This simplifies the training process considerably. Char2wav (Sotelo et al., 2017) is finetuned end-to-end in the same fashion, but requires a pre-training stage with vocoder features used for intermediate supervision.

Spectrogram prediction losses have been used extensively for feed-forward audio prediction models (Yamamoto et al., 2019; 2020; Yang et al., 2020; Arık et al., 2018; Engel et al., 2020; Wang et al.,

---

[2]This also implies that the spectrogram generation model and the vocoder have to be trained sequentially.

2019; Défossez et al., 2018). We note that the $L_1$ loss we use (along with (Défossez et al., 2018)), is comparatively simple, as spectrogram losses in the literature tend to have separate terms penalising magnitudes, log-magnitudes and phase components, each with their own scaling factors, and often across multiple resolutions. Dynamic time warping on spectrograms is a component of many speech recognition systems (Sakoe, 1971; Sakoe & Chiba, 1978), and has also been used for evaluation of TTS systems (Sailor & Patil, 2014; Chevelu et al., 2015). Cuturi & Blondel (2017) proposed the soft version of DTW we use in this work as a differentiable loss function for time series models. Kim et al. (2020) propose Monotonic Alignment Search (MAS), which relates to DTW in that both use dynamic programming to implicitly align sequences for TTS. However, they have different goals: MAS finds the optimal alignment between the text and a latent representation, whereas we use DTW to relax the constraints imposed by our spectrogram prediction loss term. Several mechanisms have been proposed to exploit monotonicity in tasks that require sequence alignment, including attention mechanisms (Graves, 2013; Zhang et al., 2018; Vasquez & Lewis, 2019; He et al., 2019; Raffel et al., 2017; Chiu & Raffel, 2018), loss functions (Graves et al., 2006; Graves, 2012) and search-based approaches (Kim et al., 2020). For TTS, incorporating this constraint has been found to help generalisation to long sequences (Battenberg et al., 2020). We incorporate monotonicity by using an interpolation mechanism, which is cheap to compute because it is not recurrent (unlike many monotonic attention mechanisms).

## 4 EVALUATION

In this section we discuss the setup and results of our empirical evaluation, describing the hyperparameter settings used for training and validating the architectural decisions and loss function components detailed in Section 2. Our primary metric used to evaluate speech quality is the Mean Opinion Score (MOS) given by human raters, computed by taking the mean of 1-5 naturalness ratings given across 1000 held-out conditioning sequences. In Appendix I we also report the Fréchet DeepSpeech Distance (FDSD), proposed by Bińkowski et al. (2020) as a speech synthesis quality metric. Appendix A reports training and evaluation hyperparameters we used for all experiments.

### 4.1 MULTI-SPEAKER DATASET

We train all models on a private dataset that consists of high-quality recordings of human speech performed by professional voice actors, and corresponding text. The voice pool consists of 69 female and male voices of North American English speakers, while the audio clips contain full sentences of lengths varying from less than 1 to 20 seconds at 24 kHz frequency. Individual voices are unevenly distributed, accounting for from 15 minutes to over 51 hours of recorded speech, totalling 260.49 hours. At training time, we sample 2 second windows from the individual clips, post-padding those shorter than 2 seconds with silence. For evaluation, we focus on the single most prolific speaker in our dataset, with all our main MOS results reported with the model conditioned on that speaker ID, but also report MOS results for each of the top four speakers using our main multi-speaker model.

### 4.2 RESULTS

In Table 1 we present quantitative results for our EATS model described in Section 2, as well as several ablations of the different model and learning signal components. The architecture and training setup of each ablation is identical to our base EATS model except in terms of the differences described by the columns in Table 1. Each ablation is "subtractive", representing the full EATS system minus one particular feature. Our main result achieved by the base multi-speaker model is a mean opinion score (MOS) of 4.083. Although it is difficult to compare directly with prior results from the literature due to dataset differences, we nonetheless include MOS results from prior works (Bińkowski et al., 2020; van den Oord et al., 2016; 2018), with MOS in the 4.2 to 4.4+ range. Compared to these prior models, which rely on aligned linguistic features, EATS uses substantially less supervision.

The **No RWDs**, **No MelSpecD**, and **No Discriminators** ablations all achieved substantially worse MOS results than our proposed model, demonstrating the importance of adversarial feedback. In particular, the **No RWDs** ablation, with an MOS of 2.526, demonstrates the importance of the raw audio feedback, and removing RWDs significantly degrades the high frequency components. **No MelSpecD** causes intermittent artifacts and distortion, and removing all discriminators results in audio that sounds robotic and distorted throughout. The **No $\mathcal{L}_{\text{length}}$** and **No $\mathcal{L}_{\text{pred}}$** ablations result in a model that does not train at all. Comparing our model with **No DTW** (MOS 3.559), the temporal

| Model | Data | Inputs | RWD | MSD | $\mathcal{L}_{\text{length}}$ | $\mathcal{L}_{\text{pred}}$ | Align | MOS |
|---|---|---|---|---|---|---|---|---|
| Natural Speech | | | | | - | | | $4.55 \pm 0.075$ |
| *GAN-TTS* (Bińkowski et al., 2020) | | | | | - | | | $4.213 \pm 0.046$ |
| *WaveNet* (van den Oord et al., 2016) | | | | | - | | | $4.41 \pm 0.069$ |
| *Par. WaveNet* (van den Oord et al., 2018) | | | | | - | | | $4.41 \pm 0.078$ |
| *Tacotron 2* (Shen et al., 2018) | | | | | - | | | $4.526 \pm 0.066$ |
| No $\mathcal{L}_{\text{length}}$ | MS | Ph | ✓ | ✓ | × | $\mathcal{L}''_{\text{pred}}$ | MI | [does not train] |
| No $\mathcal{L}_{\text{pred}}$ | MS | Ph | ✓ | ✓ | ✓ | × | MI | [does not train] |
| No Discriminators | MS | Ph | × | × | ✓ | $\mathcal{L}''_{\text{pred}}$ | MI | $1.407 \pm 0.040$ |
| No RWDs | MS | Ph | × | ✓ | ✓ | $\mathcal{L}''_{\text{pred}}$ | MI | $2.526 \pm 0.060$ |
| No Phonemes | MS | Ch | ✓ | ✓ | ✓ | $\mathcal{L}''_{\text{pred}}$ | MI | $3.423 \pm 0.073$ |
| No MelSpecD | MS | Ph | ✓ | × | ✓ | $\mathcal{L}''_{\text{pred}}$ | MI | $3.525 \pm 0.057$ |
| No Mon. Int. | MS | Ph | ✓ | ✓ | ✓ | $\mathcal{L}''_{\text{pred}}$ | Attn | $3.551 \pm 0.073$ |
| No DTW | MS | Ph | ✓ | ✓ | ✓ | $\mathcal{L}_{\text{pred}}$ | MI | $3.559 \pm 0.065$ |
| Single Speaker | SS | Ph | ✓ | ✓ | ✓ | $\mathcal{L}''_{\text{pred}}$ | MI | $3.829 \pm 0.055$ |
| EATS (Ours) | MS | Ph | ✓ | ✓ | ✓ | $\mathcal{L}''_{\text{pred}}$ | MI | $4.083 \pm 0.049$ |

Table 1: Mean Opinion Scores (MOS) for our final EATS model and the ablations described in Section 4, sorted by MOS. The middle columns indicate which components of our final model are enabled or ablated. *Data* describes the training set as Multispeaker (MS) or Single Speaker (SS). *Inputs* describes the inputs as raw characters (Ch) or phonemes (Ph) produced by Phonemizer. *RWD* (Random Window Discriminators), *MSD* (Mel-spectrogram Discriminator), and $\mathcal{L}_{\text{length}}$ (length prediction loss) indicate the presence (✓) or absence (×) of each of these training components described in Section 2. $\mathcal{L}_{\text{pred}}$ indicates which spectrogram prediction loss was used: with DTW ($\mathcal{L}''_{\text{pred}}$, Eq. 6), without DTW ($\mathcal{L}_{\text{pred}}$, Eq. 3), or absent (×). *Align* describes the architecture of the aligner as monotonic interpolation (MI) or attention-based (Attn). We also compare against recent state-of-the-art approaches from the literature which are trained on aligned linguistic features (unlike our models). Our MOS evaluation set matches that of *GAN-TTS* (Bińkowski et al., 2020) (and our "Single Speaker" training subset matches the GAN-TTS training set); the other approaches are not directly comparable due to dataset differences.

| Speaker | #1 | #2 | #3 | #4 |
|---|---|---|---|---|
| Speaking Time (Hours) | 51.68 | 31.21 | 20.68 | 10.32 |
| MOS | $4.083 \pm 0.049$ | $3.828 \pm 0.051$ | $4.149 \pm 0.045$ | $3.761 \pm 0.052$ |

Table 2: Mean Opinion Scores (MOS) for the top four speakers with the most data in our training set. All evaluations are done using our single multi-speaker EATS model.

flexibility provided by dynamic time warping significantly improves fidelity: removing it causes warbling and unnatural phoneme lengths. **No Phonemes** is trained with raw character inputs and attains MOS 3.423, due to occasional mispronunciations and unusual stress patterns. **No Mon. Int.** uses an aligner with a transformer-based attention mechanism (described in Appendix G) in place of our monotonic interpolation architecture, which turns out to generalise poorly to long utterances (yielding MOS 3.551). Finally, comparing against training with only a **Single Speaker** (MOS 3.829) shows that our EATS model benefits from a much larger multi-speaker dataset, even though MOS is evaluated only on this same single speaker on which the ablation was solely trained. Samples from each ablation are available at `https://deepmind.com/research/publications/End-to-End-Adversarial-Text-to-Speech`.

We demonstrate that the aligner learns to use the latent vector $\mathbf{z}$ to vary the predicted token lengths in Appendix H. In Table 2 we present additional MOS results from our main multi-speaker EATS model for the four most prolific speakers in our training data[3]. MOS generally improves with more training data, although the correlation is imperfect (e.g., *Speaker #3* achieves the highest MOS with only the third most training data).

---

[3] All of the MOS results in Table 1 are on samples from a single speaker, referred to as *Speaker #1* in Table 2.

## 5 DISCUSSION

We have presented an adversarial approach to text-to-speech synthesis which can learn from a relatively weak supervisory signal – normalised text or phonemes paired with corresponding speech audio. The speech generated by our proposed model matches the given conditioning texts and generalises to unobserved texts, with naturalness judged by human raters approaching state-of-the-art systems with multi-stage training pipelines or additional supervision. The proposed system described in Section 2 is efficient in both training and inference. In particular, it does not rely on autoregressive sampling or teacher forcing, avoiding issues like exposure bias (Bengio et al., 2015; Ranzato et al., 2016) and reduced parallelism at inference time, or the complexities introduced by distillation to a more efficient feed-forward model after the fact (van den Oord et al., 2018; Ping et al., 2019a).

While there remains a gap between the fidelity of the speech produced by our method and the state-of-the-art systems, we nonetheless believe that the end-to-end problem setup is a promising avenue for future advancements and research in text-to-speech. End-to-end learning enables the system as a whole to benefit from large amounts of training data, freeing models to optimise their intermediate representations for the task at hand, rather than constraining them to work with the typical bottlenecks (e.g., mel-spectrograms, aligned linguistic features) imposed by most TTS pipelines today. We see some evidence of this occurring in the comparison between our main result, trained using data from 69 speakers, against the **Single Speaker** ablation: the former is trained using roughly four times the data and synthesises more natural speech in the single voice on which the latter is trained.

Notably, our current approach does not attempt to address the text normalisation and phonemisation problems, relying on a separate, fixed system for these aspects, while a fully end-to-end TTS system could operate on unnormalised raw text. We believe that a fully data-driven approach could ultimately prevail even in this setup given sufficient training data and model capacity.

### ACKNOWLEDGMENTS

The authors would like to thank Norman Casagrande, Yutian Chen, Aidan Clark, Kazuya Kawakami, Pauline Luc, and many other colleagues at DeepMind for valuable discussions and input.

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

| Hardware | # Utt. / Batch | # Batch / Run | # Utt. / Run | Length / Utt. (s) | Length / Run (s) | Med. Run Time (s) | Realtime Factor |
|---|---|---|---|---|---|---|---|
| TPU v3 (1 chip) | 16 | 10 | 160 | 30 | 4800 | 30.53 | 157.2× |
| V100 GPU (1) | 8 | 10 | 80 | 30 | 2400 | 11.60 | 206.8× |
| Xeon CPU (6 cores) | 2 | 10 | 20 | 30 | 600 | 70.42 | 8.520× |

Table 3: EATS batched inference benchmarks, timing inference (speech generation) on a Google Cloud TPU v3 (1 chip with 2 cores), a single NVIDIA V100 GPU, or an Intel Xeon E5-1650 v4 CPU at 3.60 GHz (6 physical cores). We use a batch size of 2, 8, or 16 utterances (Utt.), each 30 seconds long (input length of 600 phoneme tokens, padded if necessary). One "run" consists of 10 consecutive forward passes at the given batch size. We perform 101 such runs and report the median run time (Med. Run Time (s)) and the resulting Realtime Factor, the ratio of the total duration of the generated speech (Length / Run (s)) to the run time. (Note: GPU benchmarking is done using single precision (IEEE FP32) floating point; switching to half precision (IEEE FP16) could yield further speedups.)

## A  HYPERPARAMETERS AND OTHER DETAILS

Our models are trained for $5 \cdot 10^5$ steps, where a single step consists of one discriminator update followed by one generator update, each using a minibatch size of 1024, with batches sampled independently in each of these two updates. Both updates are computed using the Adam optimizer (Kingma & Ba, 2015) with $\beta_1 = 0$ and $\beta_2 = 0.999$, and a learning rate of $10^{-3}$ with a cosine decay (Loshchilov & Hutter, 2017) schedule used such that the learning rate is 0 at step 500K. We apply spectral normalisation (Miyato et al., 2018) to the weights of the generator's decoder module and to the discriminators (but *not* to the generator's aligner module). Parameters are initialised orthogonally and off-diagonal orthogonal regularisation with weight $10^{-4}$ is applied to the generator, following BigGAN (Brock et al., 2018). Minibatches are split over 64 or 128 cores (32 or 64 chips) of Google Cloud TPU v3 Pods, which allows training of a single model within up to 58 hours. We use *cross-replica* BatchNorm (Ioffe & Szegedy, 2015) to compute batch statistics aggregated across all devices. Like in GAN-TTS (Bińkowski et al., 2020), our trained generator requires computation of *standing statistics* before sampling; i.e., accumulating batch norm statistics from 200 forward passes. As in GAN-TTS (Bińkowski et al., 2020) and BigGAN (Brock et al., 2018), we use an exponential moving average of the generator weights for inference, with a decay of 0.9999. Although GANs are known to exhibit stability issues sometimes, we found that EATS model training consistently converges.

Models were implemented using TensorFlow (Abadi et al., 2015) v1 framework and the Sonnet (Reynolds et al., 2017) neural network library. We used the TF-Replicator (Buchlovsky et al., 2019) library for data parallel training over TPUs.

**Inference speed.**  In Table 3 we report benchmarks for EATS batched inference on two modern hardware platforms (Google Cloud TPU v3, NVIDIA V100 GPU, Intel Xeon E5-1650 CPU). We find that EATS can generate speech two orders of magnitude faster than realtime on a GPU or TPU, demonstrating the efficiency of our feed-forward model. On the GPU, generating 2400 seconds (40 minutes) of speech (80 utterances of 30 seconds each) takes 11.60 seconds on average (median), for a realtime factor of 206.8×. On TPU we observe a realtime factor of 157.2× per chip (2 cores), or 78.6× per core. On a CPU, inference runs at 8.52× realtime.

## B    ALIGNER PSEUDOCODE

In Figure 3 we present pseudocode for the EATS aligner described in Section 2.1.

## C    SPECTROGRAM DISCRIMINATOR ARCHITECTURE

In this Appendix we present details of the architecture of the spectrogram discriminator (Section 2.3). The discriminator's inputs are $47 \times 80 \times 1$ images, produced by adding a channel dimension to the $47 \times 80$ output of the mel-spectrogram computation (Appendix D) from the length $48000$ input waveforms (2 seconds of audio at 24 kHz).

Then, the architecture is like that of the BigGAN-deep (Brock et al., 2018) discriminator for $128 \times 128$ images (listed in BigGAN (Brock et al., 2018) Appendix B, Table 7 (b)), but *removing* the first two "ResBlocks" and the "Non-Local Block" (self-attention) – rows 2-4 in the architecture table (keeping row 1, the input convolution, and rows 5+ afterwards, as is). This removes one $2 \times 2$ downsampling step as the resolution of the spectrogram inputs is smaller than the $128 \times 128$ images for which the BigGAN-deep architecture was designed. We set the channel width multiplier referenced in the table to $ch = 64$.

```python
def EATSAligner(token_sequences, token_vocab_size, lengths, speaker_ids,
                num_speakers, noise, out_offset, out_sequence_length=6000,
                sigma2=10.):
  """Returns audio-aligned features and lengths for the given input sequences.

    "N" denotes the batch size throughout the comments.

    Args:
      token_sequences: batch of token sequences indicating the ID of each token,
        padded to a fixed maximum sequence length (400 for training, 600 for
        sampling). Tokens may either correspond to raw characters or phonemes (as
        output by Phonemizer). Each sequence should begin and end with a special
        silence token (assumed to have already been added to the inputs).
        (dtype=int, shape=[N, in_sequence_length=600])
      token_vocab_size: scalar int indicating the number of tokens.
        (All values in token_sequences should be in [0, token_vocab_size).)
      lengths: indicates the true length <= in_sequence_length=600 of each
        sequence in token_sequences before padding was added.
        (dtype=int, shape=[N])
      speaker_ids: ints indicating the speaker ID.
        (dtype=int, shape=[N])
      num_speakers: scalar int indicating the number of speakers.
        (All values in speaker_ids should be in [0, num_speakers).)
      noise: 128D noise sampled from a standard isotropic Gaussian (N(0,1)).
        (dtype=float, shape=[N, 128])
      out_offset: first timestep to output. Randomly sampled for training, 0 for
        sampling.
        (dtype=int, shape=[N])
      out_sequence_length: scalar int length of the output sequence at 200 Hz.
        400 for training (2 seconds), 6000 for sampling (30 seconds).
      sigma2: scalar float temperature (sigma**2) for the softmax.

    Returns:
      aligned_features: audio-aligned features to be fed into the decoder.
        (dtype=float, shape=[N, out_sequence_length, 256])
      aligned_lengths: the predicted audio-aligned lengths.
        (dtype=float, shape=[N])
  """
  # Learn embeddings of the input tokens and speaker IDs.
  embedded_tokens = Embed(input_vocab_size=token_vocab_size,  # -> [N, 600, 256]
                          output_dim=256)(token_sequences)
  embedded_speaker_ids = Embed(input_vocab_size=num_speakers,  # -> [N, 128]
                               output_dim=128)(speaker_ids)

  # Make the "class-conditioning" inputs for class-conditional batch norm (CCBN)
  # using the embedded speaker IDs and the noise.
  ccbn_condition = Concat([embedded_speaker_ids, noise], axis=1)  # -> [N, 256]
  # Add a dummy sequence axis to ccbn_condition for broadcasting.
  ccbn_condition = ccbn_condition[:, None, :]  # -> [N, 1, 256]

  # Use `lengths` to make a mask indicating valid entries of token_sequences.
  sequence_length = token_sequences.shape[1]  # = 600
  mask = Range(sequence_length)[None, :] < lengths[:, None]  # -> [N, 600]

  # Dilated 1D convolution stack.
  # 10 blocks * 6 convs per block = 60 convolutions total.
  x = embedded_tokens
  conv_mask = mask[:, :, None]  # -> [N, 600, 1]; dummy axis for broadcast.
  for _ in range(10):
    for a, b in [(1, 2), (4, 8), (16, 32)]:
      block_inputs = x
      x = ReLU(ClassConditionalBatchNorm(x, ccbn_condition))
      x = MaskedConv1D(output_channels=256, kernel_size=3, dilation=a)(
          x, conv_mask)
      x = ReLU(ClassConditionalBatchNorm(x, ccbn_condition))
      x = MaskedConv1D(output_channels=256, kernel_size=3, dilation=b)(
          x, conv_mask)
      x += block_inputs  # -> [N, 600, 256]
  # Save dilated conv stack outputs as unaligned_features.
  unaligned_features = x  # [N, 600, 256]

  # Map to predicted token lengths.
  x = ReLU(ClassConditionalBatchNorm(x, ccbn_condition))
  x = Conv1D(output_channels=256, kernel_size=1)(x)
  x = ReLU(ClassConditionalBatchNorm(x, ccbn_condition))
  x = Conv1D(output_channels=1, kernel_size=1)(x)  # -> [N, 600, 1]
  token_lengths = ReLU(x[:, :, 0])  # -> [N, 600]
  token_ends = CumSum(token_lengths, axis=1)  # -> [N, 600]
  token_centres = token_ends - (token_lengths / 2.)  # -> [N, 600]
  # Compute predicted length as the last valid entry of token_ends.  -> [N]
  aligned_lengths = [end[length-1] for end, length in zip(token_ends, lengths)]

  # Compute output grid -> [N, out_sequence_length=6000]
  out_pos = Range(out_sequence_length)[None, :] + out_offset[:, None]
  out_pos = Cast(out_pos[:, :, None], float)  # -> [N, 6000, 1]
  diff = token_centres[:, None, :] - out_pos  # -> [N, 6000, 600]
  logits = -(diff**2 / sigma2)  # -> [N, 6000, 600]
  # Mask out invalid input locations (flip 0/1 to 1/0); add dummy output axis.
  logits_inv_mask = 1. - Cast(mask[:, None, :], float)  # -> [N, 1, 600]
  masked_logits = logits - 1e9 * logits_inv_mask  # -> [N, 6000, 600]
  weights = Softmax(masked_logits, axis=2)  # -> [N, 6000, 600]
  # Do a batch matmul (written as an einsum) to compute the aligned features.
  # aligned_features -> [N, 6000, 256]
  aligned_features = Einsum('noi,nid->nod', weights, unaligned_features)

  return aligned_features, aligned_lengths
```

Figure 3: Pseudocode for our proposed EATS aligner.

```python
import tensorflow.compat.v1 as tf

def get_mel_spectrogram(waveforms, invert_mu_law=True, mu=255.,
                        jitter=False, max_jitter_steps=60):
  """Computes mel-spectrograms for the given waveforms.

  Args:
    waveforms: a tf.Tensor corresponding to a batch of waveforms
      sampled at 24 kHz.
      (dtype=tf.float32, shape=[N, sequence_length])
    invert_mu_law: whether to apply mu-law inversion to the input waveforms.
      In EATS both the real data and generator outputs are mu-law'ed, so this is
      always set to True.
    mu: The mu value used if invert_mu_law=True (ignored otherwise).
    jitter: whether to apply random jitter to the input waveforms before
      computing spectrograms. Set to True only for GT spectrograms input to the
      prediction loss.
    max_jitter_steps: maximum number of steps by which the input waveforms are
      randomly jittered if jitter=True (ignored otherwise).

  Returns:
    A 3D tensor with spectrograms for the corresponding input waveforms.
      (dtype=tf.float32,
       shape=[N, num_frames=ceil(sequence_length/1024), num_bins=80])
  """
  waveforms.shape.assert_has_rank(2)
  t = waveforms
  if jitter:
    assert max_jitter_steps >= 0
    crop_shape = [t.shape[1]]
    t = tf.pad(t, [[0, 0], [max_jitter_steps, max_jitter_steps]])
    # Jitter independently for each batch item.
    t = tf.map_fn(lambda ti: tf.image.random_crop(ti, crop_shape), t)
  if invert_mu_law:
    t = tf.sign(t) / mu * ((1 + mu)**tf.abs(t) - 1)
  t = tf.signal.stft(t, frame_length=2048, frame_step=1024, pad_end=True)
  t = tf.abs(t)
  mel_weight_matrix = tf.signal.linear_to_mel_weight_matrix(
      num_mel_bins=80, num_spectrogram_bins=t.shape[-1],
      sample_rate=24000., lower_edge_hertz=80., upper_edge_hertz=7600.)
  t = tf.tensordot(t, mel_weight_matrix, axes=1)
  t = tf.log(1. + 10000.*t)
  return t

gen_spectrograms_for_pred_loss = get_mel_spectrogram(gen_waveforms,
                                                     jitter=False)
real_spectrograms_for_pred_loss = get_mel_spectrogram(real_waveforms,
                                                      jitter=True)
```

Figure 4: TensorFlow code for mel-spectrogram computation.

## D   MEL-SPECTROGRAM COMPUTATION

In Figure 4 we include the TensorFlow (Abadi et al., 2015) code used to compute the mel-spectrograms fed into the spectrogram discriminator (Section 2.3) and the spectrogram prediction loss (Section 2.4). Note that for use in the prediction losses $\mathcal{L}_{\text{pred}}$ or $\mathcal{L}''_{\text{pred}}$, we call this function with jitter=True for real spectrograms and jitter=False for generated spectrograms. When used for the spectrogram discriminator inputs, we do not apply jitter to either real or generated spectrograms, setting jitter=False in both cases.

```python
def soft_minimum(values, temperature):
  """Compute the soft minimum with the given temperature."""
  return -temperature * log(sum(exp(-values / temperature)))

def skew_matrix(x):
  """Skew a matrix so that the diagonals become the rows."""
  height, width = x.shape
  y = zeros(height + width - 1, width)
  for i in range(height + width - 1):
    for j in range(width):  # Shift each column j down by j steps.
      y[i, j] = x[clip(i - j, 0, height - 1), j]
  return y

def spectrogram_dtw_error(spec_a, spec_b, warp_penalty=1.0, temperature=0.01):
  """Compute DTW error given a pair of spectrograms."""
  # Compute cost matrix.
  diffs = abs(spec_a[None, :, :] - spec_b[:, None, :])
  costs = mean(diffs, axis=-1)  # pairwise L1 cost, square the diffs for L2.
  size = cost.shape[-1]

  # Initialise path costs.
  path_cost = INFINITY * ones(size + 1)
  path_cost_prev = INFINITY * ones(size + 1)
  path_cost_prev[0] = 0.0

  # Aggregate path costs from cost[0, 0] to cost[-1, -1].
  cost = skew_matrix(cost)  # Shape is now (2 * size - 1, size).
  for i in range(2 * size - 1):
    directions = [path_cost_prev[:-1],
                  path_cost[1:] + warp_penalty,
                  path_cost[:-1] + warp_penalty]
    path_cost_next = cost[i] + soft_minimum(directions, temperature)
    # Replace soft minimum with regular minimum for regular DTW.
    path_cost_next = concatenate([[INFINITY], path_cost_next])
    path_cost, path_cost_prev = path_cost_next, path_cost
  return path_cost[-1]
```

Figure 5: Pseudocode for dynamic time warping.

## E  DYNAMIC TIME WARPING PSEUDOCODE

In Figure 5 we present pseudocode for the soft dynamic time warping (DTW) procedure we use in the spectrogram prediction loss $\mathcal{L}''_{\text{pred}}$.

Note that the complexity of this implementation is quadratic. It could be made more efficient using Itakura or Sakoe-Chiba bands (Itakura, 1975; Sakoe & Chiba, 1978), but we found that enabling or disabling DTW for the prediction loss did not meaningfully affect training time, so this optimisation is not necessary in practice.

| Output symbol | x | ç | ɬ | ʲ | ; | — | ¡ | r | ~ | " |
|---|---|---|---|---|---|---|---|---|---|---|
| Substitute symbol | k | k | l | j | . | | . | | | |

Table 4: The symbols in this table are replaced or removed when they appear in phonemizer's output.

## F  TEXT PREPROCESSING

We use `phonemizer` (Bernard, 2020) (version 2.2) to perform partial normalisation and phonemisation of the input text (for all our results except for the **No Phonemes** ablation, where we use character sequences as input directly). We used the `espeak` backend (with `espeak-ng` version 1.50), which produces phoneme sequences using the International Phonetic Alphabet (IPA). We enabled the following options that phonemizer provides:

- `with_stress`, which includes primary and secondary stress marks in the output;
- `strip`, which removes spurious whitespace;
- `preserve_punctuation`, which ensures that punctuation is left unchanged. This is important because punctuation can meaningfully affect prosody.

The phoneme sequences produced by phonemizer contain some rare symbols (usually in non-English words), which we replace with more frequent symbols. The substitutions we perform are listed in Table 4. This results in a set of 51 distinct symbols. The character sequence

> *Modern text-to-speech synthesis pipelines typically involve multiple processing stages.*

becomes

> mˈɑːdən tˈɛksttəspˈiːtʃ sˈɪnθəsˌɪs pˈaɪplaɪnz tˈɪpɪkli ɪnvˈɑːlv mˌʌltɪpəl pɹˈɑːsɛsɪŋ stˈeɪdʒɨz.

## G  TRANSFORMER-BASED ATTENTION ALIGNER BASELINE

In this Appendix we describe our transformer-based attention aligner baseline, used in Section 4 to compare against our monotonic interpolation-based aligner described in Section 2.1. We use transformer attention (Vaswani et al., 2017) with output positional features as the queries, and a sum of input positional features and encoder output as the keys. The encoder outputs are from the same dilated convolution stack as used in our EATS model, normalised using Layer Normalization (Ba et al., 2016) before input into the transformer. We omit the fully-connected output layer following the attention mechanism. Both sets of positional features use the sinusoidal encodings from Vaswani et al. (2017). We use 4 heads with key and value dimensions of 64 per head. Its outputs are taken as the audio-aligned feature representations, after which we apply Batch Normalisation and ReLU non-linearity before upsampling via the decoder.

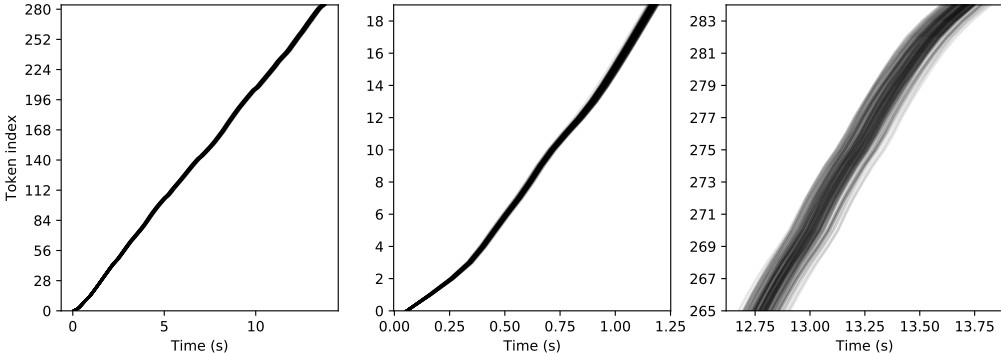

Figure 6: Positions of the tokens over time for 128 utterances generated from the same text, with different latent vectors **z**. Close-ups of the start and end of the sequence show the variability of the predicted lengths.

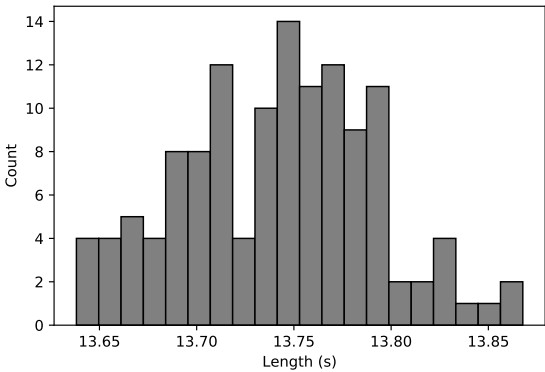

Figure 7: Histogram of lengths for 128 utterances generated from the same text, with different latent vectors **z**.

## H    VARIATION IN ALIGNMENT

To demonstrate that the aligner module makes use of the latent vector **z** to account for variations in token lengths, we generated 128 different renditions of the second sentence from the abstract: *"In this work, we take on the challenging task of learning to synthesise speech from normalised text or phonemes in an end-to-end manner, resulting in models which operate directly on character or phoneme input sequences and produce raw speech audio outputs."*. Figure 6 shows the positions of the tokens over time, with close-ups of the start and end of the sequence, to make the subtle variations in length more visible. Figure 7 shows a histogram of the lengths of the generated utterances. The variation is subtle (less than 2% for this utterance), but noticeable. Given that the training data consists of high-quality recordings of human speech performed by professional voice actors, only a modest degree of variation is to be expected.

| Model | MOS | FDSD |
|---|---|---|
| Natural Speech | $4.55 \pm 0.075$ | 0.682 |
| No Discriminators | $1.407 \pm 0.040$ | 1.594 |
| No RWDs | $2.526 \pm 0.060$ | 0.757 |
| No Phonemes | $3.423 \pm 0.073$ | 0.688 |
| No MelSpecD | $3.525 \pm 0.057$ | 0.849 |
| No Mon. Int. | $3.551 \pm 0.073$ | 0.724 |
| No DTW | $3.559 \pm 0.065$ | 0.694 |
| EATS | $4.083 \pm 0.049$ | 0.702 |

Table 5: Mean Opinion Scores (MOS) and Fréchet DeepSpeech Distances (FDSD) for our final EATS model and the ablations described in Section 4, sorted by MOS. FDSD scores presented here were computed on held-out validation multi-speaker set and therefore could not be obtained for the Single Speaker ablation. Due to dataset differences, these are also not comparable with the FDSD values reported for GAN-TTS by Bińkowski et al. (2020).

## I    EVALUATION WITH FRÉCHET DEEPSPEECH DISTANCE

We found Fréchet DeepSpeech Distances (Bińkowski et al., 2020), both conditional and unconditional, unreliable in our setting. Although they provided useful guidance at the early stages of model iteration – i.e., were able to clearly distinguish the models that do and do not train – FDSD scores of the models of reasonable quality were not in line with their Mean Opinion Scores, as shown for our ablations in Table 5.

A possible reason for FDSD working less well in our setting is the fact that our models rely on features extracted from spectrograms similar to those computed at the DeepSpeech preprocessing stage. As our models combine losses computed on raw audio *and* mel-spectrograms, it might be the case that the speech generated by some model is of lower quality, yet has convincing spectrograms. Comparison of two of our ablations seems to affirm this hypothesis: the **No MelSpecD** model achieves much higher MOS ($\approx 3.5$) than the **No RWDs** ablation ($\approx 2.5$) which is optimised only against spectrogram-based losses. Their FDSDs, however, suggest the opposite ranking of these models.

Another potential cause for the discrepancy between MOS and FDSD is the difference in samples for which these scores were established. While FDSD was computed on samples randomly held out from the training set, the MOS was computed on more challenging, often longer utterances. As we did not have ground truth audio for the latter, we could not compute FDSD for these samples. The sample sizes commonly used for the metrics based on Fréchet distance, e.g. (Heusel et al., 2017; Kurach et al., 2019; Bińkowski et al., 2020), are also usually larger than the ones used for MOS testing (van den Oord et al., 2016; Bińkowski et al., 2020); we used 5,120 samples for FDSD and 1,000 for MOS.

We also note that conditional FDSD is not immediately applicable in our setting, as it requires fixed length (two second) samples with aligned conditionings, while in our case there is no fixed alignment between the ground truth characters and audio.

We hope that future research will revisit the challenge of automatic quantitative evaluation of text-to-speech models and produce a reliable quality metric for models operating in our current regime.

| | Stages | 1 Stage | Notes |
|---|---|---|---|
| *WaveNet* (van den Oord et al., 2016) | Ling $\xrightarrow{\text{AR}}$ Au | $\times$ | |
| *SampleRNN* (Mehri et al., 2017) | $\emptyset \xrightarrow{\text{AR}}$ Au | $\times$ | not a TTS model |
| *Deep Voice* (Arik et al., 2017) | Ch $\xrightarrow{\text{AR}}$ Ph $\xrightarrow{\text{FF}}$ Ling $\xrightarrow{\text{AR}}$ Au | $\times$ | uses segmentation model |
| *WaveRNN* (Kalchbrenner et al., 2018) | Ling $\xrightarrow{\text{AR}}$ Au | $\times$ | |
| *LPCNet* (Valin & Skoglund, 2019) | Cep $\xrightarrow{\text{AR}}$ Au | $\times$ | |
| *WaveGlow* (Prenger et al., 2019) | MelS $\xrightarrow{\text{FF}}$ Au | $\times$ | |
| *FloWaveNet* (Kim et al., 2019) | MelS $\xrightarrow{\text{FF}}$ Au | $\times$ | |
| *WaveFlow* (Ping et al., 2019b) | MelS $\xrightarrow{\text{AR}}$ Au | $\times$ | partially autoregressive |
| *Par. WaveNet* (van den Oord et al., 2018) | Ling $\xrightarrow{\text{FF*}}$ Au | $\times$ | distillation |
| *ClariNet* (Ping et al., 2019a), teacher | Ch/Ph $\xrightarrow{\text{AR}}$ Au | $\checkmark$ | |
| *ClariNet* (Ping et al., 2019a), student | Ch/Ph $\xrightarrow{\text{FF*}}$ Au | $\times$ | distillation |
| *WaveGAN* (Donahue et al., 2019) | $\emptyset \xrightarrow{\text{FF}}$ Au | $\times$ | not a TTS model |
| *MelGAN* (Kumar et al., 2019) | MelS $\xrightarrow{\text{FF}}$ Au | $\times$ | |
| *Par. WaveGAN* (Yamamoto et al., 2020) | Ph $\xrightarrow{\text{AR}}$ MelS $\xrightarrow{\text{FF}}$ Au | $\times$ | |
| *AdVoc* (Neekhara et al., 2019) | MelS $\xrightarrow{\text{FF}}$ MagS | $\times$ | |
| *GAN-TTS* (Bińkowski et al., 2020) | Ling $\xrightarrow{\text{FF}}$ Au | $\times$ | |
| *Tacotron* (Wang et al., 2017) | Ch $\xrightarrow{\text{AR}}$ MelS $\xrightarrow{\text{FF}}$ MagS $\rightarrow$ Au | $\times$ | uses Griffin & Lim (1984) |
| *Tacotron 2* (Shen et al., 2018) | Ch $\xrightarrow{\text{AR}}$ MelS $\xrightarrow{\text{AR}}$ Au | $\times$ | |
| *Deep Voice 2* (Gibiansky et al., 2017) | Ch $\rightarrow$ Ph $\xrightarrow{\text{FF}}$ Ling $\xrightarrow{\text{AR}}$ Au | $\times$ | uses segmentation model |
| *DV2 Tacotron* (Gibiansky et al., 2017) | Ch $\xrightarrow{\text{AR}}$ MagS $\xrightarrow{\text{AR}}$ Au | $\times$ | |
| *Deep Voice 3* (Ping et al., 2018) | Ch $\xrightarrow{\text{AR}}$ MelS $\xrightarrow{\text{AR}}$ Au | $\times$ | several alternative vocoders |
| *TransformerTTS* (Li et al., 2019) | Ch $\rightarrow$ Ph $\xrightarrow{\text{AR}}$ MelS $\xrightarrow{\text{AR}}$ Au | $\times$ | |
| *Flowtron* (Valle et al., 2020) | Ch $\xrightarrow{\text{AR}}$ MelS $\xrightarrow{\text{FF}}$ Au | $\times$ | |
| *VoiceLoop* (Taigman et al., 2017) | Ph $\xrightarrow{\text{AR}}$ Ling $\rightarrow$ Au | $\times$ | |
| *GAN Exposure* (Guo et al., 2019) | Ph $\xrightarrow{\text{AR}}$ MelS $\xrightarrow{\text{AR}}$ Au | $\times$ | |
| *MelNet* (Vasquez & Lewis, 2019) | Ch $\xrightarrow{\text{AR}}$ MelS $\rightarrow$ Au | $\times$ | |
| *ParaNet* (Peng et al., 2019) | Ch/Ph $\xrightarrow{\text{FF*}}$ MelS $\xrightarrow{\text{FF}}$ Au | $\times$ | distillation |
| *FastSpeech* (Ren et al., 2019) | Ph $\xrightarrow{\text{FF*}}$ MelS $\xrightarrow{\text{FF}}$ Au | $\times$ | distillation |
| *Flow-TTS* (Miao et al., 2020) | Ch $\xrightarrow{\text{FF}}$ MelS $\xrightarrow{\text{FF}}$ Au | $\times$ | |
| *Glow-TTS* (Kim et al., 2020) | Ph $\xrightarrow{\text{FF}}$ MelS $\xrightarrow{\text{FF}}$ Au | $\times$ | |
| *Char2wav* (Sotelo et al., 2017) | Ch $\xrightarrow{\text{AR}}$ Ling $\xrightarrow{\text{AR}}$ Au | $\times$ | end-to-end finetuning |
| EATS (Ours) | Ch/Ph $\xrightarrow{\text{FF}}$ Au | $\checkmark$ | |

Table 6: A comparison of TTS methods. The model stages described in each paper are shown by linking together the inputs, outputs and intermediate representations that are used: characters (**Ch**), phonemes (**Ph**), mel-spectrograms (**MelS**), magnitude spectrograms (**MagS**), cepstral features (**Cep**), linguistic features (**Ling**, such as phoneme durations and fundamental frequencies, or WORLD (Morise et al., 2016) features for Char2wav (Sotelo et al., 2017) and VoiceLoop (Taigman et al., 2017)), and audio (**Au**). Arrows with various superscripts describe model components: autoregressive (**AR**), feed-forward (**FF**), or feed-forward requiring distillation (**FF\***). Arrows without a superscript indicate components that do not require learning. **1 Stage** means the model is trained in a single stage to map from unaligned text/phonemes to audio (without, e.g., distillation or separate vocoder training). EATS is the only feed-forward model that fulfills this requirement.

## J    COMPARISON OF TTS METHODS

In Table 6 we compare recent TTS approaches in terms of the inputs and outputs to each stage of the pipeline, and whether they are learnt in a single stage or multiple stages. Differentiating EATS from each prior approach is the fact that it learns a feed-forward mapping from text/phonemes to audio end-to-end in a single stage, without requiring distillation or separate vocoder training. The ClariNet teacher model (Ping et al., 2019a) is also trained in a single stage, but it uses teacher forcing to achieve this, requiring the model to be autoregressive. A separate distillation stage is necessary to obtain a feed-forward model in this case.

