# OpenReview forum: "End-to-end Adversarial Text-to-Speech"
_ICLR.cc/2021/Conference — ICLR 2021 Oral_

### Official Review · AnonReviewer2 · 2020-10-27
**Review of End-to-end adversarial text-to-speech**

**Rating:** 8
**Confidence:** 4

**Review:**


## Summary

The authors propose EATS, a method for TTS from unaligned audio and text data, directly to the waveform. Previous work either use aligned phonetic features, or output spectrograms that are later converted to a waveform by a deep vocoder model.

In order to achieve this, the authors had to use several tricks, some already existing, for instance taken from the GAN TTS architecture, and some novel. The three key novelties are:
- differentiable monotonic attention with gaussian kernel and length prediction.
- dynamic time warping for the spectrogram loss.
- using both spectrogram and waveform domain discriminators

The authors provide a comprehensive ablation study with MOS score, although their model is under the state of the art by a significant margin.

## Review

This paper builds on GAN TTS, and tries to make it trainable end-to-end without aligned features.

The two main contributions, namely dynamic time warping and monotonic attention with gaussian kernel are both elegant, and can likely be used for many other applications related to time series with heterogeneous time scales. In particular, the time warping loss allows to accomodate both for the natural irregularities in spoken speech, as well as providing sufficient signal for the monotonic attention to work.

The rest of the architecture is very similar to GAN TTS except for the spectrogram domain discriminator that was added.

While the model is under the state of the art for TTS, the samples are already quite convincing. The authors conduct a thorough ablation study, both with MOS and audio samples.

Overall I think this is a really good paper, that is likely to prove quite useful for the development of end to end speech synthesis solution. As I already mentioned, I also believe that the approach of using dynamic time warping and monotonic attention can be used for other kind of time series.


## Remarks and questions to the authors

- Table 1, MOS for Tacotron 2 would be very informative. All the baselines are trained on aligned data while Tacotron is a legitimate contender for EATS as it can be trained on the same data. The point of the authors is that their methods is simpler because the training is in one stage. However, given the large number of losses and components in their model, with their respective hyper-parameters to tune, I'm not entirely sold on the simplicity argument. The tacotron 2 paper reports a MOS of 4.5 but on a private dataset.
- Section 3, [1] used the same simple L1 + log spectrogram loss as used here.
- I was surprised by the bad performance of the transformer attention, in particular in the audio samples, the output for this model is garbage towards the end of the signal. Any clue on why this would happen?
- It would be interesting to have a benchmark, in particular, can the model generate speech in real time on GPU and on CPU?

[1] SING: Symbol-to-Instrument Neural Generator, Defossez et al. Neurips 2018.

---

> ### Author Response · Authors · 2020-11-17
> **Thank you for your comments.**
>
> Thank you for your comments.
>
> We have added Tacotron 2 to Table 1 as suggested. From our perspective the benefits of EATS relative to Tacotron 2 include the fact that EATS directly yields an efficient feed-forward speech generator (not requiring autoregressive decoding with slow inference or separate distillation to a feed-forward model), in addition to the fact that it's trained in just one stage (rather than two separate pretraining stages to separately learn the characters->mel-spectrograms and mel-spectrograms->audio mappings).
>
> Thanks for the reference to SING -- we have updated the part of our related work discussion (Sec. 3) that addresses the L1 spectrogram prediction loss to reference this work.
>
> Transformer attention's poor generalisation to longer utterances is likely due to positional encodings; the model is not trained on enough long utterances to generalise well there. The same observation was made in "Location-Relative Attention Mechanisms for Robust Long-Form Speech Synthesis" (Battenberg et al., 2020; Sec 3.4, Fig. 3).
>
> Benchmark: Appendix A in our originally submitted manuscript includes a TPU inference benchmark: batched inference with EATS generates speech at over 15x realtime per TPU v3 core (or over 30x per TPU chip). We will add a GPU benchmark in a later update.

---

> > ### Author Response · Authors · 2020-11-18
> > **Updated inference benchmarks**
> >
> > We wanted to let you know we've just updated the manuscript again to update our TPU benchmark and add a GPU benchmark. When improving our benchmarking setup we found there were large IO bottlenecks that made our original TPU benchmark artificially slow, and resolving those bottlenecks improved the measured speed by a factor of more than 5.
> >
> > In our updated benchmarks, with TPU v3 the batched inference speed is 157x realtime per chip (79x realtime per core); with a single V100 GPU, batched inference runs at 207x realtime. Please see the updated Appendix A and newly added Table 3 for additional details.

---

> > > ### Comment · AnonReviewer2 · 2020-11-18
> > > **CPU benchmark**
> > >
> > > Thanks for the update, good you managed to resolve some issues. Any chance the model can scale to real time on CPU as well? This would be quite impactful.

---

> > > > ### Author Response · Authors · 2020-11-19
> > > > **CPU benchmark**
> > > >
> > > > Thanks for the interest! We just ran a CPU benchmark; we get 8.5x realtime inference on a 6 core Intel Xeon E5-1650 v4. We've updated Appendix A again. (It's potentially an imperfect benchmark as it's running on a desktop machine sharing the CPU with other background processes etc., but we're not easily able to better isolate this.)

---

### Official Review · AnonReviewer4 · 2020-10-28
**Elegant architectural design but complicated training**

**Rating:** 7
**Confidence:** 4

**Review:**

This paper proposes a novel TTS system that 1) relies on almost no intermediate representation; and 2) is entirely feedforward instead of autoregressive. There are two major strengths in the paper. First, several modules in the proposed system are novel and smart, including the aligner and the dynamic programming loss, and it is, to my knowledge, the first feedforward text-to-speech system that does not rely on the intermediate representation. Second, the experiment result is convincing. There are, however, some room for improvement and questions for the author to clarify.

First, the elegance in the architecture is overshadowed by the complicated training algorithm. The training loss is like the superposition of common loss terms in the speech synthesis community, making the method look a bit heuristic. The complicated training algorithm also makes the proposed method harder to reproduce. It would be helpful if the authors can provide brief guidelines for readers trying to reimplement EATS, such as how to tune the hyperparameters.

Second, notice that EATS performs slightly worse than GAN-TTS, which does not quite show the benefit of end-to-end training (unlike ClariNet). I understand that there are a lot of challenges in training EATS, and the authors have briefly discussed this in Section 5. However, it is worthwhile to expand the discussion a bit by showing further experiments that demonstrate the potential benefit of end-to-end training.

Finally, although end-to-end comes with the (potential) merits of improved data efficiency and improved quality, it also has its downsides. Without a clearly interpretable hidden representation, it is harder to have direct control over prosody. How would prosody control be possible under the end-to-end framework?

Despite the weaknesses, this paper makes sufficiently novel contributions in TTS, making it above the acceptance threshold. I would look forward to further justifications of the EATS paradigm.

---

> ### Author Response · Authors · 2020-11-17
> **Thank you for your comments.**
>
> Thank you for your comments.
>
> We acknowledge that the aggregate loss function for EATS isn’t simple by any means, which is a potential downside of the method. However, through our ablation study, we have verified that each of the components in the loss function is actually necessary to achieve an acceptable MOS score of above 4 -- disabling any one single feature reduces the MOS score to 3.8 or lower, as shown in Table 1. Simplifying the loss function and the training procedure is an important goal for future work, as it would make the method more robust and easier to adapt to other datasets, speakers and/or languages.
>
> It is true that the final MOS we achieve falls short of GAN-TTS and that the main benefit of EATS relative to GAN-TTS is its end-to-end training paradigm. We believe that approaches operating in this paradigm are valuable in their potential to scale to and learn better intermediate representations from larger and more diverse TTS datasets (as we see evidence of when changing from the single speaker to multi-speaker setting), as well as for low-resource languages where language-specific tools (e.g., phonemization tools and hand-crafted language-specific text/speech features) may not be available.
>
> Regarding prosody control, if we had prosody labels available at training time, we could incorporate these into the EATS training setup in various ways. For example, if we had a single prosody label per utterance, we could use these in the same way we input speaker IDs into our speaker-conditional generator via class-conditional batch normalization. If the prosody labels were time-varying, they could be concatenated with the temporally aligned output of our proposed aligner. If you were referring to post-hoc prosody control by e.g. manipulating the latents (or intermediate representations) of a model trained without any explicit prosody inputs, it's not immediately clear how one would do this with EATS' unstructured intermediate representations. Learning a latent space with more structure that could allow for such manipulations would be an interesting future direction.

---

### Official Review · AnonReviewer3 · 2020-10-28
**A fully end-to-end text-to-speech model with limited supervision that generates speech with high quality.**

**Rating:** 8
**Confidence:** 4

**Review:**

This paper proposes a fully end-to-end TTS system with adversarial training. The proposed method has three main advantages: 1) a simple, end-to-end pipeline with only two submodules, with a fully differentiable and efficient architecture; 2) a flexible dynamic time warping to compute the loss between the predicted (generated) and true spectrograms; 3) a good MOS performance compared with the other state-of-the-art systems with more supervision. The evaluation is done very thoroughly with a variety of ablation studies.

Overall, I vote for accepting the paper. First of all, the paper is very well organized and easy to follow. Related works are well summarized, while focusing on the main differences on the proposed method. Method is explained in great detail with easy-to-follow descriptions, proper equations, and very appropriate figures. Solid evaluation is performed, and experimental results and speech samples are convincing.

Pros:
+ The structure of the paper allows an easy read.
+ The main contributions are clearly stated and supported by the experiments.
+ Major works on the similar topic are widely covered and referenced.
+ Evaluation is thorough enough to support the arguments with in-depth ablation studies.
+ Appendices provide useful, supplementary information.

Cons:
- No comparison over the computational cost nor model size is presented. It is of particular interest because the proposed model is non-autoregressive, and thus may be capable of a causal, real-time inference.
- No use of widely accepted benchmark datasets. More direct comparison would be of interest.

Minor comments/questions:
- If I understood correctly, the training sample has no phoneme-level alignment. Instead, only a sentence-speech pair is provided. If so, how do you select the corresponding text snippet from a sentence when you randomly sample 2 seconds of audio from the training examples whose length varies from 1 to 20 seconds?
- Section 2.5 on DTW is rather lengthy. DTW is a quite well-known algorithm for alignment between the two sequences, the detailed explanation on the algorithm may be omitted without the loss of readability, in my opinion.
- In Section 2.1, T is used to denote the total number of output times steps of the aligner, while T in Section 2.4 denotes the number of mel-frequency frames. Are these T's identical?
- The proposed aligner module doesn't seem to be very useful compared with the attention-based aligner as seen in the ablation study (Table 1): very small improvement from 3.551 to 3.559 MOS. Can you provide more explanations?

---

> ### Author Response · Authors · 2020-11-17
> **Thank you for your comments.**
>
> Thank you for your comments.
>
> Although we do not provide a direct comparison with other methods in terms of computational cost and model size, we do discuss the cost of inference in Appendix A. Although there was no room for this in the main paper, we have moved it there (Sec. 2) in the updated version due to relaxed space constraints. In short, with large enough batch sizes, the model generates speech an order of magnitude (15x) faster than realtime on a single TPU v3 core.
>
> Regarding the lack of phoneme-level alignment in the training data, and how EATS deals with this: we do not select the corresponding text snippet for the two second training window. Instead, the aligner always produces length predictions for the full character/phoneme sequence. We then crop out the interpolated aligner output which corresponds to the training window (which is possible because the output is now approximately aligned with the audio), and feed that to the GAN-TTS generator to produce an audio signal. The dynamic time warping loss helps to account for the fact that the generated audio signal may not be perfectly aligned with the ground truth within the selected window. As we point out at the end of Section 2.5, we still implicitly assume that the start and the end of the training window will be perfectly aligned, which is not true in general. Nevertheless, this doesn’t seem to be much of an issue in practice.
>
> Although DTW is a relatively well-known algorithm, it has not seen much use in TTS recently (except in evaluation metrics). Reiterating the algorithm also makes it easier to explain how the “soft” version differs from the original. That said, we will consider trimming down this section, or potentially moving part of it to the appendix, depending on space constraints.
>
> ‘T’ in Section 2.1 and ‘T’ in Section 2.4 both refer to the time axis, but as you point out, they have slightly different meanings (due to different resolutions). This could cause confusion. We have updated the manuscript to use a different symbol (S) in Sec. 2.1.
>
> In Table 1, we disable each component separately (in other words, the ablations are not additive). The cited MOS results of 3.551 (no monotonic interpolation) and 3.559 (no DTW) are for different ablations. For the former, DTW is enabled but monotonic interpolation is disabled; for the latter, it’s the other way around. Hence, the result without monotonic interpolation should be compared to the full EATS result at the bottom of the table (4.083, with everything enabled), indicating a clear loss of performance when disabling the monotonic aligner. This turns out to be because attention-based alignment generalises very poorly to longer utterances (the ablation samples we shared also reflect this). To avoid confusion, we have clarified in the manuscript that the ablations are not additive, but rather subtractive: each ablation represents the full EATS system, minus one particular feature. This demonstrates that all these features are necessary to achieve an MOS score above 4.

---

> > ### Author Response · Authors · 2020-11-18
> > **Updated inference benchmarks**
> >
> > We wanted to let you know we've just updated the manuscript again to update our TPU benchmark and add a GPU benchmark. When improving our benchmarking setup we found there were large IO bottlenecks that made our original TPU benchmark artificially slow, and resolving those bottlenecks improved the measured speed by a factor of more than 5.
> >
> > In our updated benchmarks, with TPU v3 the batched inference speed is 157x realtime per chip (79x realtime per core); with a single V100 GPU, batched inference runs at 207x realtime. Please see the updated Appendix A and newly added Table 3 for additional details.

---

### Official Review · AnonReviewer1 · 2020-11-02

**Rating:** 7
**Confidence:** 3

**Review:**

This paper presents an approach for end-to-end speech synthesis, where every step is learned jointly with the others. Specifically,  the proposed model takes a character sequence as input and outputs an audio signal directly. The model is trained using an combination of losses, including an adversarial loss. In the experimental section, the proposed approach is compared against strong baselines, and ablation studies are presented.

Pros:
- The proposed approach is novel and a significant step toward competitive end-to-end models.
- The related work is thorough as far as I can tell.
- The experiments are insightful, showing the impact of each part of the system.

Cons:
- The performance are promising, but still below the baselines.
- The end-to-end claim is a bit misleading as the character-based model is not performing well, and the phoneme-based model is not really end-to-end, as the g2p part is not trained jointly.
- The paper is sometime not easy to follow.

Detailed comments:
- The reason behind using an adversarial loss is not really explained in the paper. A few lines before section 2.1 would help clarify that.
- The order of the sub-sections (2.1-2.7) in Section 2 is not intuitive and seems a bit random, making the section slightly hard to follow.
- Section 2.7: "inconsistent spelling rules of the English language" -> It's not about inconsistent rules: every language has inconsistent rules, several dialects and variations in the pronunciation of words. Please rephrase.
- It's not clear which dataset was used in the experiment. If it is a private dataset, please state it clearly.

Overall, despite the paper's two main weaknesses (not fully end-to-end and lower performance),  I think it is a significant step towards fully end-to-end model and should be accepted.

---

> ### Author Response · Authors · 2020-11-17
> **Thank you for your comments.**
>
> Thank you for your comments.
>
> The phrase “end-to-end” has been used to mean a variety of different things in the context of TTS research. To clarify the intended meaning, we have included Table 5 in the appendix, which provides an overview of the different pipeline stages that are used in related work. Our work uses only one such stage, which is feed-forward and trained directly (without requiring e.g. distillation).
>
> For our best results, we do indeed use a text normalisation and phonemisation step, but nevertheless, our model does a reasonable job directly from characters (apart from the occasional mispronunciation), which is why we feel calling this paradigm “end-to-end” is justified. But we do acknowledge that the use of normalisation and phonemisation for our best results is a limitation, and note it explicitly in the last paragraph of the manuscript (Sec. 5, Discussion).
>
> We have amended Section 2 (in the text just before 2.1, as suggested) to better motivate our use of an adversarial loss function, as this runs counter to the current prevailing trend of using likelihood-based models for TTS. The use of adversarial discriminators and losses facilitates efficient feed-forward generation and inference. Also, we believe the mode seeking behaviour of adversarial losses, and the resulting focus on realism in favour of diversity, is actually an advantage in the case of TTS, because it enables the model to use its capacity more effectively. (This was already briefly mentioned in Section 3, but should also have been pointed out in Section 2.)
>
> The subsections of Section 2 are ordered to focus on the architecture and training setup first, and then the different components of the loss function are described. We'll consider reorganising this to improve readability in a later update, e.g. by dividing it explicitly into these two parts.
>
> Regarding English spelling (Section 2.7), our choice of words was perhaps unfortunate, and we have rephrased this. We meant to refer to the fact that many English phonemes can be spelled in a plethora of different ways (e.g. [f] in fish, laugh, half, graph, sapphire), and many graphemes can correspond to multiple different phonemes depending on context (e.g. <ough> in though, tough, through, thorough). Many languages have orthographies that are much more regular than English in this respect (e.g. Spanish, Turkish, Finnish), but admittedly no human language is probably fully regular in this regard.
>
> We have also updated Section 4.1 of the manuscript to state clearly that a private dataset was used for this work.

---

### Decision · Program_Chairs · 2021-01-07
**Final Decision**

**Decision:**

Accept (Oral)

**Comment:**

This paper investigates a speech synthesis approach that directly generates raw audios from text or phoneme inputs in an end-to-end fashion.  The approach first maps the input texts/phonemes into a representation sequence that is aligned with the output at a lower sampling frequency by a differentiable aligner and then upsamples the representation sequence to the full audio frequency by a decoder.  A number of techniques including adversarial training and soft DTW are applied to improve the training.  The experimental results are good. There are raised concerns from the reviewers which are mostly cleared by the rebuttal of the authors.  After the rebuttal and discussion, all reviewers are supportive on accepting the paper.